# First Case of *Candida auris* Colonization in a Preterm, Extremely Low-Birth-Weight Newborn after Vaginal Delivery

**DOI:** 10.3390/jof7080649

**Published:** 2021-08-10

**Authors:** Alessio Mesini, Carolina Saffioti, Marcello Mariani, Angelo Florio, Chiara Medici, Andrea Moscatelli, Elio Castagnola

**Affiliations:** 1Istituto di Ricerca e Cura a Carattere Scientifico (IRCCS) Istituto Giannina Gaslini, Largo G. Gaslini, 5, 16147 Genova, Italy; alessiomesini@gaslini.org (A.M.); carolinasaffioti@gaslini.org (C.S.); marcellomariani@gaslini.org (M.M.); chiaramedici@gaslini.org (C.M.); andreamoscatelli@gaslini.org (A.M.); 2Department of Neuroscience, Rehabilitation, Ophthalmology, Genetics, and Maternal and Child Sciences (DINOGMI), University of Genoa, 16128 Genova, Italy; angeloflorio91@libero.it

**Keywords:** *Candida auris*, extremely low birth weight neonate, preterm neonate, *Candida* colonization, antifungal prophylaxis

## Abstract

*Candida auris* is a multidrug-resistant, difficult-to-eradicate pathogen that can colonize patients and health-care environments and cause severe infections and nosocomial outbreaks, especially in intensive care units. We observed an extremely low-birth-weight (800 g), preterm neonate born from vaginal delivery from a *C. auris* colonized mother, who was colonized by *C. auris* within a few hours after birth. We could not discriminate whether the colonization route was the birth canal or the intensive care unit environment. The infant died on her third day of life because of complications related to prematurity, without signs or symptoms of infections. In contexts with high rates of *C.auris* colonization, antifungal prophylaxis in low-birth-weight, preterm neonates with micafungin should be considered over fluconazole due to the *C. auris* resistance profile, at least until its presence is excluded.

## 1. Background

*Candida* bloodstream infections (BSIs) are the third most common cause of health care-associated BSIs, especially in extremely low-birth-weight (<1000 g) and/or preterm (<35 weeks gestational age) neonates [1].

Epidemiology of *Candida* spp. has changed worldwide in recent years, with emergence of resistant non-*albicans* strains. From its first identification in 2009 in Japan [2], *Candida auris* has raised global attention due to its rapid spread and its multidrug-resistance profile [3], with resistance to all three major classes of antifungals (azoles, echinocandins, and polyenes) [4,5]. Fluconazole resistance is observed frequently in all four *C. auris* clades, and hence its use in the treatment of or prophylaxis in specific settings is limited [6]. In the past, *C. auris* isolates were most often misidentified as a range of other *Candida* species, especially as *C. haemulonii*, to whom it is phylogenetically closely related [7,8]; however, the use of matrix-assisted laser desorption/ionization time-of-flight mass spectrometry (MALDI-TOF MS) can differentiate *C. auris* from other *Candida* species. Molecular methods based on DNA sequencing can also identify *C. auris*; sequencing of the D1-D2 region of the 28s ribosomal DNA (rDNA) or the internal transcribed spacer (ITS) region of rDNA are accepted methods. More recently, the development of polymerase chain reaction (PCR) assays specific for *C. auris* and for *C. auris*-related species using cultured colonies has shown promise for the rapid and accurate identification of *C. auris*, which could prove particularly useful in outbreak situations [9].

*C. auris* frequently causes outbreaks, especially in intensive care units (ICU), including neonatal intensive care units (NICU). *C. auris* is normally identified as a colonizer, more rarely as a cause of BSI or deep site localisations, with horizontal transmission probably as the most frequent way of acquisition [10], since carriers represent an important reservoir. Continuous carriage for up to three months after initial isolation of *C. auris* has been documented [11]. Notably, despite risk factors for *C. auris* invasive disease being substantially identical to those for other *Candida* species (e.g., medical procedures and devices including central venous line, urinary catheters, surgery, broad spectrum antibiotics exposure, ICU admission) [12], colonization represents one of the main risks for invasive infection [7,8]. Prevention strategies such as isolation precautions, 1:1 nurse to patient ratios, and prophylaxis with micafungin instead of azoles can reduce incidence and mortality due to *C. auris* [13].

The aim of this report is to describe the first case of possible vertical transmission of *C. auris* and its management, and to discuss the possibility of using prophylaxis with micafungin in contexts of high risk of azole-resistant fungi nosocomial acquisition.

## 2. Case Report

An extremely low weight birth (800 g), preterm (25 weeks of gestational age), female neonate was born from vaginal delivery following preterm labor complicated by maternal bleeding in *abruptio placentae* in another Center. The mother was admitted to the ICU because of COVID-19 pneumonia necessitating intubation and mechanical ventilation. Notably, vaginal swab of the mother resulted positive for *C. auris*.

The newborn was transferred within a few hours after delivery to the NICU of Istituto Giannina-Gaslini, Genoa, Italy. Multiple swabs for *C. auris* were performed at time of admission following the CDC’s procedures for high-risk patients [10]. Cultures of the axilla, skin, eyes and ears resulted positive. Yeast identification was carried out by MALDI-TOF MS (bioMérieux) using the manufacturer’s validated library database and the antifungals sensitivity test by broth microdilution Sensititre YeastOne ITAMYUCC (Thermo Scientific, Waltham, MA, USA). Currently, no specific breakpoints are established for *C. auris*, so the results interpretation was defined according to CDC’s tentative breakpoints [10]: fluconazole: > 256 µg/µL; amphotericin B: 1 µg/µL; caspofungin: 0.12 µg/µL; anidulafungin: 0.25 µg/µL; micafungin:0.12 µg/µL; voriconazole: 2 µg/µL; posaconazole: 0.12 µg/µL; itraconazole: 2 µg/µL (Table 1). Minimal inhibitory concentrations (MIC) results were overlapping to those of the strain isolated from the mother. According to the hospital protocol, fluconazole prophylaxis was started before yeast, Moreover, even before *C. auris* identification, strict contact isolation procedures were implemented according to the hospital policy for patients coming from other Centers to contain the spread of resistant pathogens [14,15]. The infant died on her third day of life because of the complications related to the extreme prematurity and the severe hypoxic–ischemic encephalopathy with worsening of cerebral function monitoring pattern and complete absence of electric cerebral activity. No signs and symptoms of invasive fungal infection were observed. During the neonate stay in the NICU, no horizontal spread of *C. auris* was observed.

## 3. Comments

This is the first description of *C. auris* vertical transmission, since, to the best of our knowledge, no case of colonization of a neonate born from vaginal delivery has been reported (MEDLINE and Scholar search using keywords: “*Candida auris*”, “mother to child”, “vaginal transmission”, “vertical transmission”, last check on 17 July 2021). The patient was colonized within a few hours after delivery, confirming the possibility of colonization by resistant pathogens already in the first hours of life, as observed in other contexts [16]. However, in this case, we can not completely exclude that *C. auris* was acquired from the environment and hands of healthcare workers, although acquisition through the birth canal from a colonized mother was more likely.

Remarkably, the introduction of *C. auris* in Italy was a recent event and its spread could have been facilitated by the COVID-19 pandemic, as was probably the case for our patient. Indeed, a recent a study conducted in the same hospital where the patient was delivered showed the spread of multi-drug resistant (MDR) Gram-negatives and *C. auris* during the SARS-CoV-2 pandemic [17]. The close genetic relatedness of *C. auris* strains observed in the patient and her mother was consistent with nosocomial transmission of the pathogen, suggesting that protective measures used to prevent SARS-CoV-2 acquisition do not share the benefit of preventing cross-transmission of other pathogens between patients. Therefore, dedicated strategies are warranted to prevent horizontal spread and maintain effective antimicrobial stewardship programs in the setting of COVID-19 care, particularly in the case of *C. auris* due to the difficulty of its eradication and the very limited therapeutic options once infection has ensued.

Although further studies and more evidence is needed to understand the risk of vertical transmission, according to the recent recommendation issued by the International Society for Antimicrobial Chemotherapy Working Group on Infection Prevention and Control [18], we suggest that patients transferred from a healthcare environment with high prevalence of *C. auris* infection/colonization should be isolated and should routinely be investigated for *C. auris* colonization because of the risk of BSI and its poor outcomes, and because of the risk of horizontal spread. Fluconazole prophylaxis is considered a standard for low-birth-weight neonates [19]. However, because of the high prevalence of fluconazole-resistance in *C. auris*, micafungin, with less than 10% of resistant strains described [20], should be considered instead of fluconazole as prophylaxis regimen in very low-birth-weight neonates admitted in or coming from hospitals with high prevalence of *C. auris*, at least until its presence is excluded.

## 4. Conclusions

We presented a brief case of a low-birth-weight preterm female neonate colonized by *C. auris*, possibly acquired from the birth canal. Invasive candidiasis prophylaxis was performed with fluconazole, as per our hospital protocol, before we identified azole-resistant *C. auris* in surveillance cultures.

In light of the rising prevalence of *C. auris* in hospitals, micafungin prophylaxis for invasive candidiasis should be considered as an alternative to fluconazole in low-birth-weight neonates.

## Figures and Tables

**Table 1 jof-07-00649-t001:** Comparison between CDC-suggested tentative MIC breakpoints and antifungal susceptibility test from patient (and mother) isolated strains.

Drug	CDC Tentative MIC Breakpoints (µg/mL)	MIC from Patient Strain (µg/mL)
Fluconazole	≥32	>256
Voriconazole	N/A	2
Posaconazole	N/A	0.12
Itraconazole	N/A	2
Amphotericin B	≥ 2	1
Anidulafungin	≥ 4	0.25
Caspofungin	≥ 2	0.12
Micafungin	≥ 4	0.12

## Data Availability

Anonymized clinical data are available at request.

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
