# Peer review of "First Case of Candida auris Colonization in a Preterm, Extremely Low-Birth-Weight Newborn after Vaginal Delivery"

_jof, 2021, doi:10.3390/jof7080649_

Round 1

Reviewer 1 Report

Many thanks for this case. This adds to the existing literature on possible acquisition of the MDR C auras from the birth canal via SVD.

Some comments to the authors.

  1. Abstract. birth weight and outcome of the baby could be added in the abstract to inform the readers
  2. The aim of this report could be added as the last sentence in the introduction
  3. Case presentation. Please remove the citation on SARS CoV 2 in the case description section , transfer it to the comment/discussion section if its really necessary  
  4. The comment/discussion section lacks a conclusion section- to summarise the case and provide ways forward

Author Response

Dear Sir,

thanks for comments and suggestions.

The report was reviewed by an English- speaking colleague to improve meaning of periods.

As you requested, we included sex of neonate in the case report. We commented about Covid pandemic and spread of MDR bacteria and fungi, reporting a recent study of the major adult hospital of our city (the mother of our neonate patient was admitted in ICU of this hospital).

Reviewer 2 Report

Overall impression

The authors present a brief case report of a low birthweight preterm infant colonized with C. auris, an emerging fungal pathogen. The infant was prophylactically treated with the antifungal fluconazole, but later an antifungal sensitivity test revealed the organism to be azole resistant. The authors conclude that in light of the rising prevalence of azole resistant C. auris in hospitals, micafungin treatment should be considered as an alternative prophylactic treatment for candidiasis. The report is concise, comprehensive and the included comments are supported by evidence. The manuscript present a finding that could prevent patient harm and is worthy of publication.

Detailed points

Case Report section: Please include the sex of the infant at the start of the case. While not highly relevant in the newborn, Candida species respond to sex hormones and have higher virulence in females.

Comments: Line 84 – please comment on how the COVID pandemic has facilitated the spread of C. auris. While there is some evidence of opportunistic Candida Sp. respiratory tract infections secondary to COVID, in this case the mother presented with Candida vaginitis and not Candida pneumonia. Do the authors suggest that crowded hospitals with overburdened staff facilitate the spread of infections in general and Candidiasis in particular?  

Writing

The grammar needs work. E.g. the meaning of line 95 is not immediately clear.

Author Response

Dear Sir,

thanks for comments and suggestions.

The report was reviewed by an English- speaking colleague.

As you requested, we included weight and outcome in the abstract section and we added the aim of the report in the introduction section. We removed SARS-COV 2 citation from case description and we expressed our opinion about COVID pandemic and spread of MDR bacteria and fungi in comment section (as requested by rev 1).

Finally we created a discussion section focusing on key points of the report.

Reviewer 3 Report

The paper reports the first case of Candida auris colonization in a preterm newborn.  

The paper is well written but i have some comments:

  • the introduction must be improved
  • Add the profile of resistence, report the value of MIC  in a table 
  • the referenche must be improved

Author Response

thanks for comments and suggestions.

 The report was reviewed by an English- speaking colleague.

We improved introduction section and we added a new reference. As you requested, we summarized antifungals and respective MICs in a table.